# Socioeconomic, Eating- and Health-Related Limitations of Food Consumption among Polish Women 60+ Years: The ‘ABC of Healthy Eating’ Project

**DOI:** 10.3390/nu14010051

**Published:** 2021-12-23

**Authors:** Jadwiga Hamulka, Joanna Frackiewicz, Beata Stasiewicz, Marta Jeruszka-Bielak, Anna Piotrowska, Teresa Leszczynska, Ewa Niedzwiedzka, Anna Brzozowska, Lidia Wadolowska

**Affiliations:** 1Department of Human Nutrition, Institute of Human Nutrition Sciences, Warsaw University of Life Sciences (WULS-SGGW), Nowoursynowska 159C, 02-776 Warsaw, Poland; marta_jeruszka_bielak@sggw.edu.pl (M.J.-B.); anna_brzozowska@sggw.edu.pl (A.B.); 2Department of Human Nutrition, Faculty of Food Science, University of Warmia and Mazury in Olsztyn, Sloneczna 45F, 10-718 Olsztyn, Poland; beata.stasiewicz@uwm.edu.pl (B.S.); ewa.niedzwiedzka@uwm.edu.pl (E.N.); lidia.wadolowska@uwm.edu.pl (L.W.); 3Department of Functional and Organic Food, Institute of Human Nutrition Sciences, Warsaw University of Life Sciences (SGGW-WULS), Nowoursynowska 159C, 02-776 Warsaw, Poland; anna_piotrowska@sggw.edu.pl; 4Department of Human Nutrition and Dietetics, Faculty of Food Technology, University of Agriculture in Krakow, Balicka 122, 30-149 Krakow, Poland; teresa.leszczynska@urk.edu.pl

**Keywords:** older women, food consumption, socioeconomic status, eating limitations, health limitations, mini nutritional assessment (MNA^®^), anthropometric data, handgrip strength

## Abstract

The study aimed at identifying the socioeconomic, eating- and health-related limitations and their associations with food consumption among Polish women 60+ years old. Data on the frequency of consumption of fruit, vegetables, dairy, meat, poultry, fish, legumes, eggs, water and beverages industrially unsweetened were collected with the Mini Nutritional Assessment (MNA^®^) and were expressed in the number of servings consumed per day or week. Three indexes: the Socioeconomic Status Index (SESI), the Eating-related Limitations Score (E-LS) and the Health-related Limitations Score (H-LS) were developed and applied. SESI was created on the base of two variables: place of residence and the self-reported economic situation of household. E-LS included: difficulties with self-feeding, decrease in food intake due to digestive problems, chewing or swallowing difficulties, loss of appetite, decrease in the feeling the taste of food, and feeling satiety, whereas H-LS included: physical function, comorbidity, cognitive function, psychological stress and selected anthropometric measurements. A logistic regression analysis was performed to assess the socioeconomic, eating-, and health-related limitations of food consumption. Lower socioeconomic status (vs. higher) was associated with a lower chance of consuming fruit/vegetables ≥ 2 servings/day (OR = 0.25) or consuming dairy ≥ 1 serving/day (OR = 0.32). The existence of multiple E-LS limitations (vs. few) was associated with a lower chance of consuming fruit/vegetables ≥ 2 servings/day (OR = 0.72), consuming dairy ≥ 1 serving/day (OR = 0.55) or consuming water and beverages industrially unsweetened ≥6 cups/day (OR = 0.56). The existence of multiple H-LS limitations was associated with a lower chance of consuming fruit/vegetables ≥ 2 servings/day (OR = 0.79 per 1 H-LS point increase) or consuming dairy ≥ 1 serving/day (OR = 0.80 per 1 H-LS point increase). Limitations found in the studied women were related to insufficient consumption of selected groups of food, which can lead to malnutrition and dehydration. There is a need for food policy actions, including practical educational activities, to eliminate barriers in food consumption, and in turn to improve the nutritional and health status of older women.

## 1. Introduction

The world’s population is aging and the elderly population are the fastest-growing segment of the world population. This is specifically true for women who live longer than men. The number of people aged over 60 years is estimated to more than double, from 900 million (12%) in 2015 to 2 billion (22%) in 2050 worldwide [1,2]. Similarly, in Poland, the percentage of people aged 60 years and over is still growing. In 2015, people at this age represented about 16% of the Polish population, while it is expected to constitute 23% in 2030, with a higher share of women [3]. The consequence of an aging society is the occurrence of various health problems in this group, which are associated with higher morbidity, mortality, and medical costs [4,5]. Older adults suffer from chronic diseases such as diabetes, heart disease, hypertension, age-related macular degeneration, and cancers, as well as specific geriatric conditions such as frailty and falls, cognitive impairments and gum diseases [6]. It is often the result of an unhealthy lifestyle, including poor nutrition and low physical activity. Furthermore, eating patterns in aging population are influenced by multiple factors including age, living environment, socioeconomic, psychological, and medical determinants.

Available studies show that the older adults have inadequate consumption of fruit, vegetables, dairy, fish, legumes, and water. A diet low in these foods may be characterized by low nutritional density, low content of many nutrients, and bioactive components, which may cause general malnutrition [7,8], which in turn results in worsening the physical and mental health and in lowering the overall quality of life [9,10]. The causes of the above-mentioned dietary inadequacies might be comorbidities, as well as the physiological and psychological state of the elderly population, which in many ways affect the consumption of foods [11,12]. In addition, socioeconomic status, place of residence, and family habits in dietary intake can also contribute to food consumption and nutritional status by a variety of available foods [13,14].

Ageing population often suffers from gum diseases, tooth loss, decay and mouth infections, e.g., xerostomia [15]. In turn, those impairments may impact food perception by changing texture perception and the release of flavour components, which have a significant impact on food acceptability [16]. Moreover, the changes in the perception of hunger and diminished pleasure associated with food consumption result in appetite loss. It has often been suggested that older adults who lost their sense of taste may eat less food or choose stronger flavours. In addition, senile changes in the perception of taste are also important, e.g., stronger sweet and salty stimuli are needed [17,18].

Insufficient hydration is one of the most important age-related problems [19], and the prevalence of dehydration among older adults has been estimated to reach 20–30% [20]. There is increasing evidence that dehydration, even at moderate level, plays an important role in the development of various morbidities. Current findings suggest that it can cause constipation, impaired cognitive function, falling, orthostatic hypotension, salivary dysfunction, and poor control of hyperglycaemia in diabetes or hyperthermia [21,22].

Identifying risk factors for malnutrition including limitations on the consumption of particular food groups is of utmost important to better guide intervention and prevention strategies in food and health policies for older populations. Recent research has been focused mainly on identifying dietary patterns in various contexts [23,24,25]. Hence, to fill the gap in research on multifactorial limitations of food consumption among older people, there was a need for a comprehensive look at this problem, which takes into account a number of aspects regarding the functioning, eating and health factors of elderly population. For these reasons, three indexes: the Socioeconomic Status Index (SESI), the Eating-related Limitations Score (E-LS) and the Health-related Limitations Score (H-LS) were developed and applied. SESI was created on the base of two variables: place of residence and the self-reported economic situation of household. E-LS included: difficulties with self-feeding, decrease in food intake due to digestive problems, chewing or swallowing difficulties, loss of appetite, decrease in the feeling the taste of food, and feeling satiety, whereas H-LS included: physical function, comorbidity, cognitive function, psychological stress and selected anthropometric measurements. Thus, the objective of this study was to identify the socioeconomic, eating- and health-related limitations and their associations with food consumption among Polish women 60+ years old.

## 2. Materials and Methods

### 2.1. Ethics Approval

The project followed the ethical standards recognized by the Declaration of Helsinki and was approved by the Bioethics Committee of the Faculty of Medical Sciences, University of Warmia and Mazury in Olsztyn on 17 June 2010 (Resolution No. 20/2010). All participants provided their written informed consent to take participation in the study.

### 2.2. Study Design and Participants

This study was designed as a cross-sectional. Data were collected in 2015 as part of the national multicentre ‘ABC of Healthy Eating’ project (1st edition). The project was conducted by well-trained academic researchers from 7 Polish universities in 8 locations covering the entire territory of Poland, including urban, sub-urban and rural areas.

Recruitment was carried out through press advertisements, seniors’ houses of daily living, communal centres and rural housewives’ circles, universities of the 3rd century and researchers’ personal contact. Respondents 60+ years old were invited to attend. The recruitment was directed to a larger number of respondents with a lower social or economic status, but no strict criteria were established. We realized this assumption by recruiting in various places and limiting the participation of affluent older people, specifically with higher education, who are generally more willing to take part in such projects. Due to the recruitment procedure applied, we achieved the group with SES quite typical for Polish older women (Statistics Poland, 2021) [26].

The main inclusion criteria were as follows: (i) age: ≥60 years, (ii) no communication problems, (iii) location up to 50 km from the academic centres and (iv) interest in participation in the education program and written consent to participate in the study (Figure 1).

In the study, 418 respondents across Poland were recruited. Then, 57 participants were excluded from analyses because they were less than 60 years old and there was a lack of data on socioeconomic status or food consumption (Figure 1). The initial sample included 361 respondents aged 60+ years of both sexes. Due to the low rate of men, gender differences in food consumption and anthropometric measurements, and the lack of some anthropometric data, and 48 men (11% of the recruited sample) were excluded from the study [27,28]. Finally, the present study consisted of 313 Polish women aged 60–89 years. Details on the total sample characteristics are given in the Results section.

### 2.3. Dietary Data

Dietary data were collected with the Mini Nutritional Assessment (MNA^®^) [29,30], the Simplified Nutritional Appetite Questionnaire (SNAQ), and the food frequency questionnaire (FFQ) based on the Habits and Nutrition Beliefs Questionnaire (KomPAN^®^) [31,32]. The questionnaires were self-administered by respondents and supervised by researchers, who explained orally any doubts or filled in the questionnaires when problems with reading/writing occurred. The MNA^®^ included the usual consumption of five food groups: (i) fruit/vegetables as the main dietary source of dietary fiber, vitamins, and minerals, (ii) dairy, (iii) meat/poultry/fish and (iv) legumes/eggs as markers of protein intake, linked to the risk of malnutrition, as well as (v) water and beverages industrially unsweetened as sources of water, linked to the risk of dehydration [21,29]. The recall period of food consumption was the previous 12 months. The cut-offs for consumption of fruit/vegetables, dairy, meat/poultry/fish, and legumes/eggs were based on the dichotomous data expressed in the number of servings/day or week according to the Mini Nutritional Assessment (MNA^®^) [29]. The consumption of water and beverages industrially unsweetened was expressed in the number of cups/day (MNA^®^), and then was converted into two categories based on the distribution and the average consumption (approx. 1400 mL/day). Finally, the food consumption was expressed in the number of servings consumed per day or week as follows:

fruit or vegetables <2, ≥2 servings/day (serving size: 80–100 g, e.g., medium-sized tomato or apple, 1 cup of raw leafy greens, ½ cup of cut-up fruit or vegetables, 1 cup of 100% fruit or vegetable juice);dairy (milk, fermented milk drinks, cheese, etc.) <1, ≥1 serving/day (serving size: e.g., a cup of milk, buttermilk, kefir or yoghurt, 100 g of cottage cheese, 2 slices of cheese);meat or poultry or fish <1, ≥1 serving/day (serving size: 100–200 g, e.g., 100 g of meat, 2 slices of ham, 84 g of cooked fish or seafood);legumes (bean soup, pea soup, cooked beans etc.) or eggs <2, ≥2 servings/week (serving size: e.g., ¼ cup of cooked beans or peas, 1 egg or 2 egg whites);water and beverages industrially unsweetened (water, juice, coffee, tea, etc., excluding sweetened beverages coca-cola type) <6, ≥6 cups/day. Such categories were created due to the distribution and the average consumption of water and beverages industrially unsweetened in this study (approx. 1400 mL/day).

A lower food consumption was considered when the consumption of food groups was as follows: <2 servings/day of fruit/vegetables, <1 serving/day of dairy, <1 serving/day of meat/poultry/fish, <2 servings/week of legumes/eggs, and <6 cups/day of water and beverages industrially unsweetened. On the other hand, as a higher food consumption, we considered the following cut-offs: ≥2 servings/day of fruit/vegetables, ≥1 serving/day of dairy, ≥1 serving/day of meat/poultry/fish, ≥2 servings/week of legumes/eggs, and ≥6 cups/day of water and beverages industrially unsweetened.

### 2.4. Anthropometric Measurements

The complete data of anthropometric measurements including body weight (W; kg), height (H; cm), waist circumference (WC; cm), and strength of the right and left arm muscles (kg) were collected for 264 participants. All anthropometric measurements were proceeded by qualified researchers and according to the standardized procedures. Professional equipment and measuring tape were used, the same type across all the research centres. Measurements were taken in light clothing and without shoes twice, and average values were calculated [33].

Weight was measured using the electronic digital scale to the nearest 0.1 kg (SECA 799, Hamburg, Germany). Height was measured with a portable stadiometer with the head in the horizontal Frankfurt plane and recorded with a precision of 0.1 cm (SECA 220, Hamburg, Germany). Body mass index (BMI) was calculated according to the formula: W (kg)/H (m^2^) and was categorized according to the WHO standards for adults [34]; as for elderly population, there is a lack of clear cut-offs. BMI 18.5–24.9 kg/m^2^ was interpreted as normal weight, BMI 25.0–29.9 kg/m^2^ was interpreted as overweight and BMI ≥ 30.0 kg/m^2^ as obesity. Waist circumference was measured with a stretch-resistant tape that provides constant 100 g tension (SECA 201, Hamburg, Germany), at the midway point between the iliac crest and the costal margin (lower rib) on the anterior axillary line in a resting expiratory position. Handgrip strength (HGS; kg) was measured with maximal effort, using a hydraulic hand dynamometer with adjustable widths (manufacturer; SAEHAN Corporation, Masan, Korea) with a precision of 0.5 kg [35]. Each volunteer was asked to squeeze the dynamometer two times with each hand. To control the effect of fatigue, approximately 2-min rest was applied between each measurement. Cut-point used for low HGS in women was ≤20 kg [35].

### 2.5. Socioeconomic Status Index (SESI)

The Socioeconomic Status Index (SESI) was created on the base of two variables: (i) place of residence, and (ii) the self-reported economic situation of household—factors that clearly affect health and life expectancy based on evidence from large European studies [7,36,37,38]. Considering the place of residence, the points were given according to data from Statistics Poland [26], which indicated a relationship between place of residence and income, namely higher economic status and living conditions of city dwellers in relation to towns and villages. The SESI was calculated by summing the scores assigned to each category of components mentioned above, and then was expressed in the range from 0 to 6 points (Table 1). The higher SESI score indicates a lower socioeconomic status.

According to the SESI’s distribution and the median value of 2.0, the socioeconomic status was categorized at three levels: ‘higher’ (0–1 point), ‘average’ (2 points), and ‘lower’ (3–6 points).

### 2.6. Eating-Related Limitations Score (E-LS)

Eating-related Limitations Score (E-LS) is an original proposal of the score of limitations that includes: (i) difficulties with self-feeding, (ii) decrease in food intake due to digestive problems, chewing or swallowing difficulties, (iii) loss of appetite, (iv) decrease in the feeling the taste of food, and (v) feeling satiety, associated with the physiological aging-related changes [16,17,18,39,40]. The E-LS was calculated based on the sum of points that were assigned to each category of five components (Table 2). Data related to the E-LS’ components were collected using the MNA^®^ [29] and the Simplified Nutritional Appetite Questionnaire (SNAQ) [41], as two validated tools to screen the risk of malnutrition in ageing population. Some of the E-LS’s components were recoded when compared to the original version for easier interpretation. Cronbach’s alpha was calculated, and its value (0.466) indicated quite good internal consistency of the E-LS. The E-LS was expressed in the range from 0 to 7 points, and the higher number of points indicates more limitations related to eating. According to the E-LS’s distribution and the median value of 2.0, the E-LS was set at two levels: ‘lower’ (≤2 points) and ‘higher’ (3–7 points).

### 2.7. Health-Related Limitations Score (H-LS)

The Health-related Limitations Score (H-LS) is an original proposal of the score of limitations which included: physical function, comorbidity, cognitive function, and psychological stress, widely used in the studies associated with the health condition of older adults [6,36,42,43,44]. The H-LS was calculated based on the sum of points assigned to each category of 13 components (Table 3). Data related to the H-LS’ components were collected using the MNA^®^ [29] and measurements of waist circumference and strength of the right and left arm muscles (see Section 2.4. Anthropometric measurements), which are widely used in the assessment of the nutritional status of elderly population [34,45]. The sex-specific cut-off point of the waist circumference was used according to the WHO recommendation based on the evidence considering the increased risk of chronic diseases and mortality in large cohort studies of ageing population [46]. The cut-off points of the strength of arm muscles were established according to the European Working Group on Sarcopenia in Older People—EWGSOP [35]. Some of the H-LS’s components were recoded in comparison with the original version for easier interpretation. The internal consistency of the H-LS was measured with the Cronbach’s alpha and equaled 0.541. The H-LS was expressed in the range from 0 to 13 points, and the higher number of points indicates more limitations related to the health condition. According to the H-LS’s distribution and the median value of 4.0, the H-LS was established at two levels: ‘lower’ (< 4 points) and ‘higher’ (4–13 points).

### 2.8. Statistical Analysis

Data were presented as a sample percentage (%) for categorical data or mean and standard deviation (SD) for continuous data. The differences between groups were verified with the Pearson Chi^2^ test (categorical data) or the Kruskal–Wallis test (continuous data; for more than two groups) or the Tukey’ test (continuous data; for two groups). Before statistical analysis, the normality of variable distribution was checked with a Kolmogorov–Smirnov test. The logistic regression analysis was performed to assess a chance of higher food consumption in association with an average and lower socioeconomic status, or higher level of limitations related to eating or health conditions, as well as with a one-point increase in SESI, E-LS, and H-LS. The odds ratios (ORs) and 95% confidence intervals (95% CI) were calculated. The lower food consumption, higher socioeconomic status, or lower level of eating- or health-related limitations were used as reference (ref.). ORs were adjusted for age (continuous variable in years) and SESI (continuous variable in points), if applicable. The level of significance of the OR was verified with the Wald’s test [47]. For all tests, *p* < 0.05 was considered significant. The statistical analysis was performed using STATISTICA software version 12.0 (StatSoft Inc., Tulsa, OK, USA; StatSoft, Krakow, Poland).

## 3. Results

The study sample characteristics are shown in Table 4. The total sample included 313 women aged 60.0–89.0 (69.5 ± 5.6) years. Based on the SESI, 31% of women had lower socioeconomic status. According to E-LS and H-LS, more limitations related to eating and health conditions were found in 43% and 51% of women, respectively.

### 3.1. Distribution of the Components of Socioeconomic, Eating- and Health-Related Limitations Scores

The distribution of the components of SESI is presented in Figure 2. Most of the women lived in the cities above 100 thousand inhabitants due to their easier access to academic centres conducting the project. Moreover, almost half of the sample declared to live thriftily and very thriftily, and poorly.

Taking into account the components of the E-LS (Figure 3), it was shown that more than half of women declared that their feeling of both the food taste and appetite were as good as other people at their age.

According to components of H-LS, the majority of women had excessive body weight, evaluated their nutritional status as ‘good’ and had waist circumference ≥88 cm (Figure 4). In our study, neither women with BMI < 18.5 kg/m^2^ nor self-reported as malnourished were found.

### 3.2. Associations of Socioeconomic Status, Eating- and Health-Related Limitations with Food Consumption

The SESI, E-LS, and H-LS values and the consumption of particular food groups among women 60+ years are presented in Table 5, whereas detailed data on food groups consumption in relation to the single components of these indexes are shown in Appendix A. Lower consumption of fruit/vegetables or dairy was more frequent in women with lower than higher SESI. Higher means of E-LS were associated with lower consumption of fruit/vegetables, dairy, legumes/eggs as well as with water and beverages industrially unsweetened. Higher consumption of dairy or water and beverages industrially unsweetened was more frequent in women with lower than higher E-LS. Higher means of H-LS were related to lower consumption of fruit/vegetables, dairy or water and beverages industrially unsweetened.

Results of logistic regression analysis are presented in Figure 5a–e while detailed data for selected components of the E-LS and H-LS are presented in Appendix A. Lower socioeconomic status (ref. higher) was associated with the lower adherence to the higher consumption (ref. lower) of fruit/vegetables by 75% (Figure 5a) or with the lower adherence to the higher consumption (ref. lower) of dairy by 68% (Figure 5b). One-point increase in socioeconomic status was associated with the lower adherence to the higher consumption (ref. lower) of fruit/vegetables by 22% (Figure 5a) or with the lower adherence to the higher consumption (ref. lower) of dairy by 29% (Figure 5b). Higher E-LS (ref. lower E-LS) was associated with the lower adherence to the higher consumption (ref. lower) of dairy by 45% (Figure 5b) or with the lower adherence to the higher consumption (ref. lower) of water and beverages industrially unsweetened by 44% (Figure 5e). One-point increase in E-LS was associated with the lower adherence to the higher consumption (ref. lower) of fruit/vegetables by 28% (Figure 5a) or with the lower adherence to the higher consumption (ref. lower) of water and beverages industrially unsweetened by 24% (Figure 5e). One-point increase in H-LS was associated with the lower adherence to the higher consumption (ref. lower) of fruit/vegetables by 21% (Figure 5a) or with the lower adherence to the higher consumption (ref. lower) of dairy by 20% (Figure 5b).

After logistic regression analysis was performed, no significant associations were found between SESI, E-LS, H-LS and the consumption of meat/poultry/fish (Figure 5c) or the consumption of legumes/eggs (Figure 5d).

## 4. Discussion

This study provides interesting insights regarding socioeconomic, eating- and health-related limitations of food consumption among women over 60 years old. To the best of our knowledge, this study was the first to examine a comprehensive assessment of food consumption limitations using three scores, of which three were newly created as an original authors’ proposition. While there are studies reporting the dietary patterns in elderly population [23,24,25], direct findings related to limitations on the consumption of particular food groups are lacking. This study demonstrates that the low socioeconomic status was associated with lower consumption of fruit/vegetables or dairy. Eating-related limitations were associated with lower consumption of fruit/vegetables, dairy or water and beverages industrially unsweetened. Health-related limitations were associated with lower consumption of fruit/vegetables or dairy.

### 4.1. Socioeconomic Status and Food Consumption

Our findings indicate that low socioeconomic status was associated with lower consumption of fruit/vegetables. Such association between fruit/vegetables consumption and socioeconomic status measured with various variables has been reported in many studies for different populations [48,49,50], also for older adults [51,52], although not in all [7,53]. Interestingly, in our study, women living in villages or towns when compared to cities’ residents had higher intake of fruit/vegetables, probably due to the possible self-supply from own gardens and orchards. On the other hand, in the rural areas, a higher distance to supermarkets may result in a lower consumption of fruit and vegetables [54].

In the present study, low socioeconomic status was also associated with the lower dairy consumption. To our knowledge, in the last decade, there is little research on the association between socioeconomic factors and dairy consumption in the elder population. Moreover, in the available studies conducted among adults, the results are inconsistent. In Brazilian adults, a high socioeconomic status, including the high family income, was positively associated with the amount of yoghurt consumption [55], whereas, in young American adults, socioeconomic status had no effect on the amount of dairy consumption calculated in total or separately for milk, cheese and yogurt [56]. On the contrary, as Kapaj and Deci [57] indicated in their review book, socioeconomic characteristics influenced the consumption of dairy products all over the world.

In Poland, in the period prior to our study, the prices of milk and its products had increased greater than those of other foods, which may explain our finding at least partially [58]. Moreover, the dairy product sector has recently undergone the greatest change, which involved, among others, the introduction of new functional products not always accepted by consumers in older age [59]. We can also speculate that the subjects of low socioeconomic status in our study lacked the knowledge about the dietary guidelines for the consumption of fruit/vegetables and dairy as low socioeconomic status is generally associated with a low level of education and low nutrition knowledge [60].

### 4.2. Eating-Related Limitations of Food Consumption

Our results indicate that a higher score of the Eating-related limitations index, and its single components, like the decrease in food consumption and worse feeling of both food taste and appetite, were associated with lower consumption of the fruit/vegetables. Moreover, the weaker appetite was also associated with the lower consumption of legumes/eggs or water and beverages industrially unsweetened. Limitations on the consumption of fruit/vegetables, especially raw ones in the older adults, may be due to the occurrence of gum disease and tooth loss. This has an effect on chewing ability and chewing foods, which in turn changes the perception of texture, a significant factor of food acceptance. Additionally, in older people, the dislike of bitter and sour flavours, typical for many vegetables and fruit, is more pronounced [17]. The diminished enjoyment of consumption of fruit/vegetables may influence the decision to limit these foods in older persons [15,16]. A study conducted among older Canadian women showed that a greater feeling of hunger was connected with generally higher diet quality, including consumption of vegetables and fruit [61]. The authors explained the sensation of hunger driven for food consumption, and possibly healthier food choices among the elderly population. Moreover, better self-evaluated appetite was associated with more positive nutrition-related attitudes, including declarations of more conscious food choices and behaviours among older adults from five European countries in the NU-AGE project [60]. It is known that loss of appetite, as well as low food consumption and chewing problems, can be associated with the development of malnutrition and dehydration, and generally lower quality of life, especially in the ageing population [62].

### 4.3. Health-Related Limitations of Food Consumption

We have found that more health-related limitations in general, as well as some components like weaker self-evaluated health status and nutritional status self-assessed as ‘does not know’, were associated also with lower consumption of fruit/vegetables among Polish female elders. Those associations may be bidirectional, namely, eating less vegetables and fruit results in worse health status, or just the opposite—existence of some health problems is a cause of decreased consumption of these foods. Södergren et al. [63] found that each additional daily serving of fruit and vegetables was associated with higher odds of reporting health as good or better in 3644 Australian women and men aged 55–65 years. In addition, there are many studies showing a positive effect of the higher consumption of fruit and vegetables on the prevention of chronic diseases and on the reduced risk of mortality in older adults [4,15,64].

In this study, the nutritional status self-assessed as ‘does not know’ was associated with lower consumption of dairy, which may result in lower animal protein and calcium intake. As far as we know, there is little research analysing the factors that may increase the dairy consumption among older persons. The studies mainly concern the association between dairy consumption and the risk of metabolic diseases or bone health [65,66,67]. Intake of two-three servings of dairy per day, particularly products low in fat and fermented (like yogurts and kefirs), offer benefits with regard to overall health, and specifically bone health, which is extremely important in elderly women [68,69].

### 4.4. Strengths and Limitations

The main strength of the current study is a relatively homogeneous group of women aged 60+ years from the multicentre study that was a part of a nationwide project ‘ABC of Healthy Eating’. Although the sample was not randomly selected, it covers the entire territory of Poland and widely reflects the demographic-social diversity of Poles, thus forming a good basis for generalizations. Next, a comprehensive and innovative study approach should be highlighted. We applied the self-assessment of the economic situation, health, and nutritional status and also three newly developed indexes: the SESI, the E-LS, and the H-LS, aimed at identifying risk factors for malnutrition including limitations on the consumption of particular food groups. Moreover, taking into account the low cost, convenience of conducting research as well as a good ability to predict nutritional status, nutritional screening assessment tools can contribute to the early diagnosis of people with greater malnutrition risk. Hence, the indicators developed and used in our study may be useful for the health protection and physical condition of the ageing population not only in Poland, but also in other countries with similar living conditions.

The findings of this study should be also considered within the context of its limitations. First, due to the cross-sectional design of the study, the data for each respondent were obtained at one point in the study; therefore, only the factor-outcome relationship can be considered, and no causality can be pointed. Secondly, the sample size is relatively small and non-representative at the population-level; however, it was calculated in regard to the main objective of the project. Adequacy of sample size was checked for data under the study and the post-hoc statistical power was calculated. For example, when means of the Eating-Associated Limitations score (3.1 ± 1.4 vs. 2.4 ± 1.4 points; 47 vs. 266 subjects) or the Health-Conditions Limitations Score (4.0 ± 2.2 vs. 2.9 ± 1.9 points; 47 vs. 266 subjects) for two groups were compared, and a 5% significance level was assumed, the statistical power was 88.5% or 89.7%, respectively. Thus, based on the calculations and considering the precision of methods used, we have found that the sample size was sufficient to detect differences between groups. Next, all indexes, including the socioeconomic status index, were constructed using mainly subjects’ self-reported data, like the economic situation of household, which may be biased due to the subjective estimate. On the other hand, a rigid calculation of income per family member is not always adequate for the amount of money spent ‘on themselves’ by elders in the family. It was shown that the self-reported economic situation of the household often better reflects the actual level of satisfaction with the socio-economic quality of life in older adults [38]. Lastly, although the diet is a complex matrix of various foods and nutrients, only five food groups were included in the food consumption assessment. However, for good reason, analyses focused on the consumption of fruit/vegetables, dairy, meat/poultry/fish, legumes/eggs, water and beverages industrially unsweetened, due to their close link with the nutritional and hydration status among the elderly [21,29]. This selection, as well as the limitation of the portion size, was limited by the MNA^®^ used to obtain data in this study. Nevertheless, MNA^®^ is a validated and widely used questionnaire for older adults [29,30].

## 5. Conclusions

Our study revealed that the main limitations on the consumption of selected food groups among Polish elderly women were: lower socioeconomic status, worse taste, lower food consumption in terms of quantity, worse health condition reported by the respondents and weaker appetite. Interestingly, the most limitations were detected mainly for fruits/vegetables followed by dairy, and water and beverages industrially unsweetened. Insufficient consumption of these food groups can result in a poor-quality diet. This, in turn, can lead to malnutrition and dehydration, and further to disability and loss of independence of older adults, given the effect of a vicious circle through increased food consumption limitations associated with both functional and health conditions. A lot of additional research is needed to determine the food consumption limitations, along with the quantitative assessment of food consumption among older people, including men. Moreover, understanding the elderly’s socioeconomic and health situation is important to formulate prevention plans adequate for this population’s reality.

Our findings suggest a need for the implementation of new strategies of food policy activities aimed at eliminating barriers of food consumption for improving the nutritional and health status and thus improving the quality of life of older adults as a vulnerable subpopulation. The nutritional programs should focus on practical aspects like adoption of healthier dietary practices into real life, e.g., the improving cooking skills to improve appetite. Such initiatives could be introduced at a country or local level. The priority should be vegetables and fruit, and their consumption should reach at least five portions a day or account for half of the “plate” according to Polish guidelines [70] and the Planetary Health Diet developed by EAT-Lancet Commission [71]. The activities should reinforce not only more conscious and proper food choices in order to improve health, but also have a positive impact on the environment. This can be a part of a global solution to today’s global nutritional challenges [72].

## Figures and Tables

**Figure 1 nutrients-14-00051-f001:**
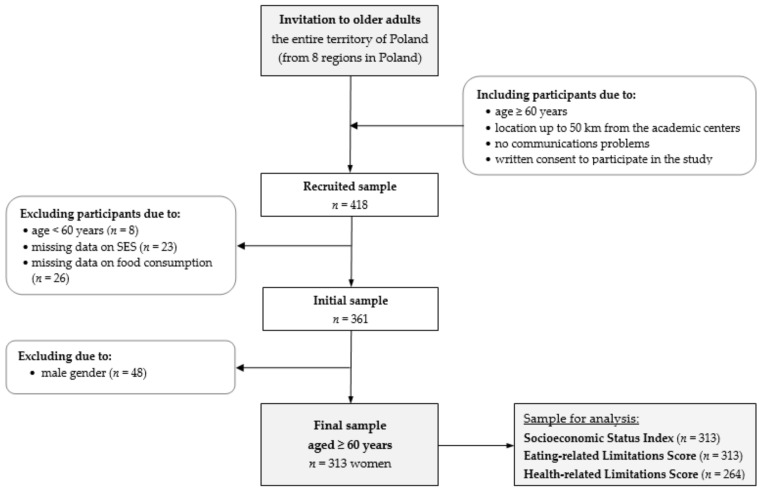
Flowchart of study design and sample collection. Notes: SES—socioeconomic status.

**Figure 2 nutrients-14-00051-f002:**
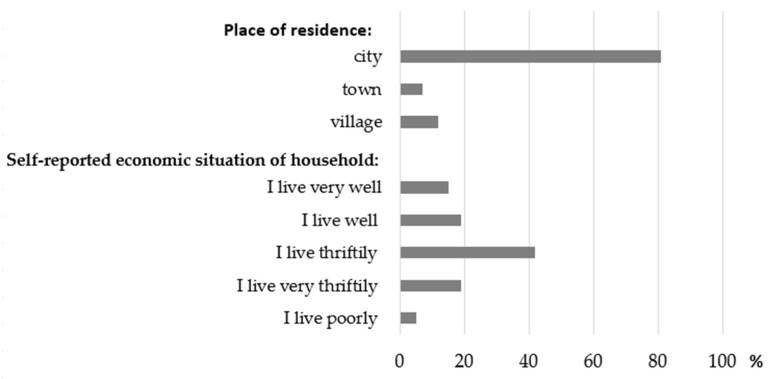
Distribution of the components of the Socioeconomic Status Index (percentage of the total sample).

**Figure 3 nutrients-14-00051-f003:**
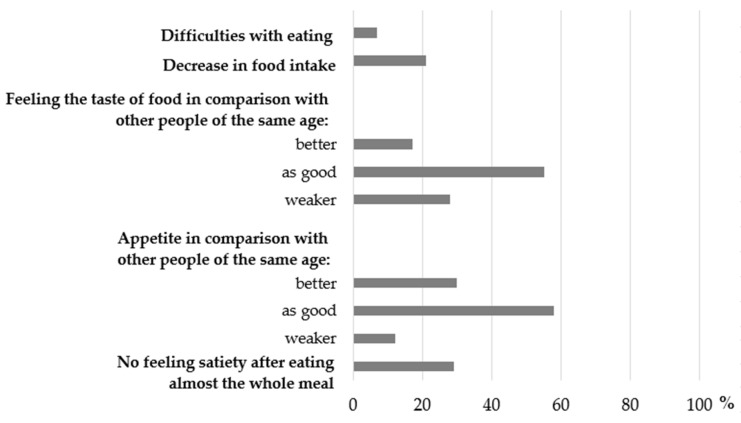
Distribution of the components of the Eating-related Limitations Score (percentage of the total sample).

**Figure 4 nutrients-14-00051-f004:**
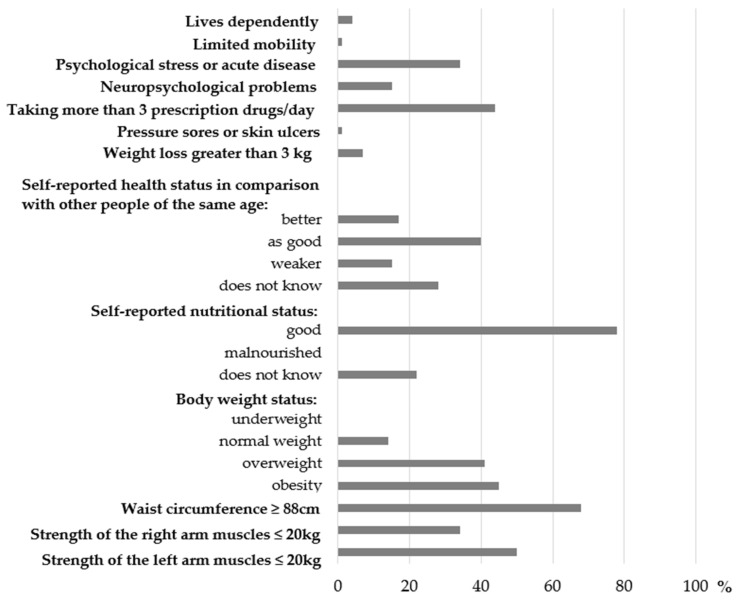
Distribution of the components of the Health-related Limitations Score (percentage of the total sample).

**Figure 5 nutrients-14-00051-f005:**
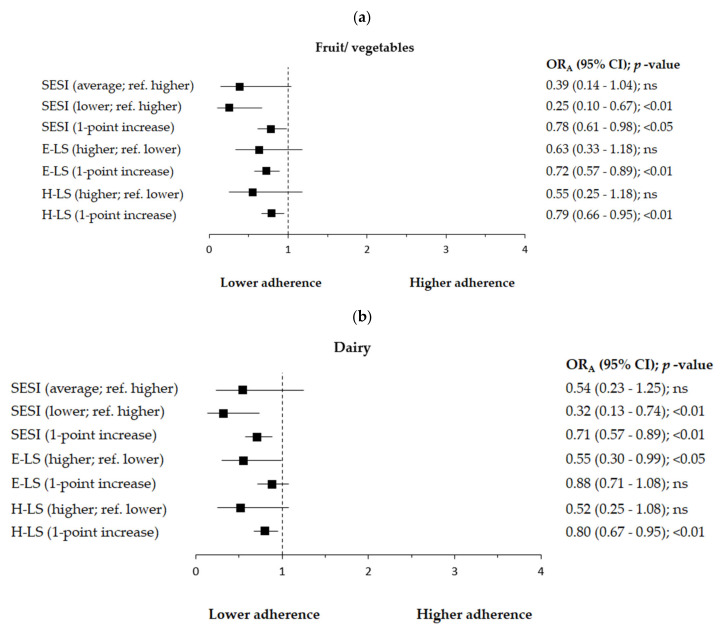
(**a**–**e**). Factors associated with higher consumption of (**a**) fruit/vegetables (≥2 servings/day vs. <2 servings/day as reference), (**b**) dairy (≥1 serving/day vs. < 1 serving/day as reference), (**c**) meat/poultry/fish (≥1 serving/day vs. <1 serving/day as reference), (**d**) legumes/eggs (≥2 servings/week vs. <2 servings/week as reference), and (**e**) water and beverages industrially unsweetened (≥6 cups/day vs. <6 cups/day as reference) among Polish women aged 60+. Notes: SESI—Socioeconomic Status Index; E-LS—Eating-related Limitations Score; H-LS—Health-related Limitations Score; OR_A_—odds ratio adjusted for age (continuous variable in years) and socioeconomic status score (continuous variable in points) excluding the modelled variable from confounders set, respectively; ref.—the reference category; 95% CI—95% Confidence Interval; *p*-value—level of significance assessed by Wald’s test; ns—statistically insignificant.

**Table 1 nutrients-14-00051-t001:** The Socioeconomic Status Index (SESI)—components data and scoring.

Components	Scoring (Points)
Place of residence:	
city (>100,000 inhabitants)	0
town (<100,000 inhabitants)	1
village	2
Self-reported economic situation of household:	
I live very well—I have enough resources for everything, and I put off savings	0
I live well—I have enough resources for everything, but I don’t put off savings	1
I live thriftily—I have enough resources for everything	2
I live very thriftily—I have enough resources only for basic needs (food/clothing/housing fees)	3
I live poorly—I don’t have enough resources even for basic needs (food/clothing/housing fees)	4

Range of points: 0–6

**Table 2 nutrients-14-00051-t002:** Eating-related Limitations Score (E-LS)—components data and scoring.

Components	Scoring (Points)
Difficulties with self-feeding (self-fed with some difficulty or unable to eat without assistance)	
no	0
yes	1
Decrease in food intake in the last 3 months (due to loss of appetite, digestive problems, chewing or swallowing difficulties)	
no	0
yes	1
Feeling the taste of food in comparison with other people of the same age	
better	0
as good	1
weaker	2
Appetite in comparison with other people of the same age	
better	0
as good	1
weaker	2
Feeling satiety after eating almost the whole meal	
yes	0
no	1

Range of points: 0–7

**Table 3 nutrients-14-00051-t003:** The Health-related Limitations Score (H-LS)—components data and scoring.

Components	Cut-Offs for Scoring (Points)	
	0	1
Lives dependently (i.e., in nursing home or hospital)	no	yes
Limited mobility (i.e., able to get out of bed/chair but does not go out vs. goes out)	no	yes
Psychological stress or acute disease in the last 3 months	no	yes
Neuropsychological problems	no	yes
Taking more than 3 prescription medications/day	no	yes
Pressure sores or skin ulcers	no	yes
Weight loss greater than 3 kg in the last 3 months	no	yes
Self-reported health status in comparison with other people of the same age (i.e., weaker/does not know vs. as good/better)	no	yes
Self-reported nutritional status (malnourished/does not know vs. good)	no	yes
BMI (kg/m^2^)	18.5–29.9	<18.5 or ≥30
Waist circumference (cm)	<88	≥88
Strength of the right arm muscles (kg)	>20	≤20
Strength of the left arm muscles (kg)	>20	≤20

Range of points: 0–13

**Table 4 nutrients-14-00051-t004:** Study sample characteristics.

Variables	Sample Percentage (%) or Mean ± SD
Sample size	313
Age, years	69.5 ± 5.6
60–69	63
70–89	37
BMI (kg/m^2^)	29.8 ± 4.8
Waist circumference (cm)	94.0 ± 11.2
Strength of the right arm muscles (kg)	22.5 ± 6.0
Strength of the left arm muscles (kg)	20.5 ± 5.5
Socioeconomic Status Index (SESI) ^a^, points	2.1 ± 1.3
higher, 0–1	27
average, 2	42
lower, 3–6	31
Eating-related Limitations Score (E-LS) ^b^, points	2.5 ± 1.4
lower, ≤2	57
higher, 3–7	43
Health-related Limitations Score (H-LS) ^c^, points	3.6 ± 2.0
lower, <4	49
higher, 4–13	51
Consumption of:	
Fruit/vegetables, servings/day	
lower, <2	15
higher, ≥2	85
Dairy, servings/day	
lower, <1	18
higher, ≥1	82
Meat/poultry/fish, servings/day	
lower, <1	23
higher, ≥1	77
Legumes/eggs, servings/week	
lower, <2	34
higher, ≥2	66
Water and beverages industrially unsweetened ^d^, cups/day	
lower, <6	53
higher, ≥6	47

Notes: data for BMI (Body Mass Index), waist circumference, strength of the arms muscles and the Health-related Limitations Score (H-LS) were obtained for 264 women; ^a^ calculated based on the place of residence and self-declared economic situation of household; ^b^ calculated based on the: difficulties with self-feeding, decrease in food intake in the last 3 months, feeling the taste of food in comparison with other people of the same age, appetite in comparison with other people of the same age, and feeling satiety after eating almost the whole meal; ^c^ calculated based on the: lives dependently, limited mobility, psychological stress or acute disease in the last 3 months, neuropsychological problems, taking more than 3 prescription drugs/day, pressure sores or skin ulcers, weight loss greater than 3 kg in the last 3 months, self-reported health status in comparison with other people of the same age, self-reported nutritional status, BMI, waist circumference, strength of the right and left arms muscles; ^d^ water, juice, coffee, tea, etc., excluding sweetened beverages coca-cola type.

**Table 5 nutrients-14-00051-t005:** The socioeconomic, eating, and health-related limitations scores of food consumption among Polish women 60+ years (% of the sample or mean ± SD).

Variables	Consumption of														
	Fruit/Vegetables (Servings/Day)			Dairy (Servings/Day)			Meat/Poultry/Fish (Servings/Day)			Legumes/Eggs (Servings/Week)			Water and Beverages Industrially Unsweetened ^d^ (Cups/Day)		
	<2	≥2	*p*	<1	≥1	*p*	<1	≥1	*p*	<2	≥2	*p*	<6	≥6	*p*
Sample size	47/34 ^#^	266/230 ^#^		57/41 ^#^	256/223 ^#^		72/60 ^#^	241/204 ^#^		105/90 ^#^	208/174 ^#^		167/134 ^#^	146/130 ^#^	
Socioeconomic Status Index (SESI) ^a^, points	2.5 ± 1.1	2.0 ± 1.3	<0.01	2.6 ± 1.4	2.0 ± 1.2	<0.01	2.1 ± 1.2	2.1 ± 1.3	ns	2.2 ± 1.2	2.1 ± 1.3	ns	2.2 ± 1.4	2.1 ± 1.2	ns
higher, 0–1	13	30		16	30		26	28		23	30		29	26	
average, 2	40	42	<0.05	40	42	<0.05	44	41	ns	40	42	ns	38	46	ns
lower, 3–6	47	28		44	28		29	32		37	28		34	28	
Eating-related Limitations Score (E-LS) ^b^, points lower, ≤2 higher, 3–7	3.1 ± 1.4	2.4 ± 1.4	<0.01	2.8 ± 1.4	2.4 ± 1.4	<0.05	2.5 ± 1.4	2.5 ± 1.4	ns	2.7 ± 1.3	2.4 ± 1.4	<0.05	2.7 ± 1.4	2.2 ± 1.4	<0.001
	47	59	ns	44	60	<0.05	57	57	ns	50	61	ns	50	65	<0.01
	53	41		56	40		43	43		50	39		50	35	
Health-related Limitations Score (H-LS) ^c^, points lower, <4 higher, 4–13	4.6 ± 2.1	3.5 ± 2.0	<0.01	4.6 ± 2.4	3.4 ± 1.9	<0.01	3.5 ± 2.0	3.7 ± 2.1	ns	3.9 ± 2.2	3.5 ± 2.0	ns	3.9 ± 2.1	3.4 ± 2.0	<0.05
	35	51	ns	34	52	<0.05	50	49	ns	46	51	ns	45	54	ns
	65	49		66	48		50	51		54	49		55	46	

Notes: #data for Health-related Limitations Score (H-LS); ^a^ calculated based on the place of residence and self-declared economic situation of household; ^b^ calculated based on the: difficulties with self-feeding, decrease in food intake in the last 3 months, feeling the taste of food in comparison with other people of the same age, appetite in comparison with other people of the same age, and feeling satiety after eating almost the whole meal; ^c^ calculated based on the: lives dependently, limited mobility, psychological stress or acute disease in the last 3 months, neuropsychological problems, taking more than 3 prescription drugs/day, pressure sores or skin ulcers, weight loss greater than 3 kg in the last 3 months, self-reported health status in comparison with other people of the same age, self-reported nutritional status, BMI, waist circumference, strength of the right and left arms muscles; ^d^ water, juice, coffee, tea, etc., excluding sweetened beverages coca-cola type; *p*—the level of significance was assessed by Kruskal–Wallis test (continuous variables) or chi^2^ test (categorical variables).

## Data Availability

Due to ethical restrictions and participant confidentiality, data cannot be made publicly available. However, data from the ABC of Healthy Eating study are available upon request, for researchers who meet the criteria for access to confidential data. Data requests can be sent to ABC of Healthy Eating study coordinator (Jadwiga Hamulka).

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
