# Peer review of "Socioeconomic, Eating- and Health-Related Limitations of Food Consumption among Polish Women 60+ Years: The ‘ABC of Healthy Eating’ Project"

_nutrients, 2021, doi:10.3390/nu14010051_

Round 1

Reviewer 1 Report

Thank you for the opportunity to review this manuscript about a Polish sample of older women and their eating behaviours. I thought this was a valuable piece of research in terms of the sample and area of interest but have some questions about the methodology used (please see specific comments below). I also think the Results section should be much more concise. There are countless figures and the text only serves to repeat much of what is included in the graphs a lot of the time. I recommend the authors make a decision about whether they want to represent the results graphically (my preference) or in the text and make substantial cuts to the alternative. If the figures represent the data, there is no need to repeat all the information in the text, just pull out the key findings.

Abstract

Generally well written with some grammatical errors.

Unclear throughout what is meant by ‘limitations’. This should be clarified from the start to allow the reader to understand what the paper’s objectives are.

Introduction

As with abstract a few grammatical errors, but otherwise well written with good rationale.

I think it would be helpful towards the end to be a bit more specific with the objectives of the study to give the reader a better understanding of what the paper is about. This relates to my earlier point about not really understanding what is meant by ‘limitations’ at this point.

Method

Grammatical errors in this section are more substantial and I recommend a professional proof reader.

How was the recruitment directed to a lower SES with no criteria? This needs more transparency.

More justification needed for excluding men. 48 is still a large enough number that it would be interesting to look at.

Sample characteristics would conventionally be presented in the Methods, alongside the ‘Participants’ section.

Typo in Figure 1 which reads ‘duo’ and should read ‘due’.

Please could you give some clearer details about how the food frequency data was collected? You state that the researchers were supervising, but I’m unclear how long the recall period was, and how they ‘supervised’ (i.e. did they ask the questions orally or did they just watch while people filled out the questionnaire?).

It would be helpful to explain how the cut-off was decided for the ‘lower’ and ‘higher’ food consumption scores. Were they arbitrary or based on any literature?

More context is needed as to why living in the city is higher SES than a town which is higher than a village. This is not true at all of many other countries and given that this is an international journal, if this is specifically true of Poland, this needs to be stated and supported by a reference. Alternatively, if it cannot be justified I would recommend reconsidering how participants were allocated their SES.

On a related note, was the self-reported SES status based on a validated measure?

Section 2.6 is the first time I start to understand properly what is meant by ‘limitations’. This could be much more clearly explained in the Introduction with more examples and explanations.

Table 3. Unclear what happened to anyone with BMI < 18.5? Was there anyone? If so, what did you do with these participants in terms of scoring?

Section 3.1 (l.277 onwards) – revisit the point about whether this is an appropriate way to categorise SES? If it is, was it not inevitable that there would be such a skew with this method of recruitment?

As a general point, the results are quite long because information is often repeated in the text and the figures. The paper could be written more concisely if the figures were designed more clearly (i.e. label the axis) and could speak for themselves to a certain extent.

Figures 5 – 9 could also be clearer, both in terms of what it is representing and the graphic.

Is there a way that some of the figures could be presented together? The Results section is exceptionally long at the moment in a way that I don’t think it needs to be.

Discussion

As above, some attention to the language could improve clarity.

The conclusions end with a point about Covid-19 which has not been mentioned before. The authors should consider whether this is a) important (in which case I recommend mentioning earlier) or b) not important (in which case I recommend removing).

Author Response

Dear Reviewer,

We are excited to re-submit the improved and changed version of our manuscript with a new title:    “Socioeconomic, Eating- and Health-Related Limitations of Food Consumption among Polish Women 60+ Years. The ‘ABC of Healthy Eating’ Project”.             

We really appreciate all the comments from Reviewer, since they helped us to improve our paper.
We have addressed all issues indicated in the review report, and believe that the new version will meet the journal publication requirements. Thank you very much for a thorough review and insightful feedback. We agree with suggestions and tried to address them accordingly. Please find our responses in the file attached.

Looking forward to hearing from you,

Yours sincerely,

Manuscript authors

Reviewer 2 Report

This is a very interesting study regarding the main limitations on the consumption of selected food groups among Polish elderly women, namely lower socioeconomic status, higher diet-related limitations (worse taste, lower food consumption, weaker appetite) and worse health condition.

Author Response

Dear Reviewer,

We are excited to re-submit the improved and changed version of our manuscript with a new title:    “Socioeconomic, Eating- and Health-Related Limitations of Food Consumption among Polish Women 60+ Years. The ‘ABC of Healthy Eating’ Project”.             

We thank you for positive feedback.

Yours sincerely,

Manuscript authors
